# AT WHICH TRAINING STAGE DOES CODE DATA HELP LLMS REASONING?

**Yingwei Ma**[1,2]*, **Yue Liu**[1]*, **Yue Yu**[1,2]†, **Yuanliang Zhang**[1], **Yu Jiang**[3], **Changjian Wang**[1], **Shanshan Li**[1]†

[1]National University of Defense Technology
[2]Peng Cheng Laboratory
[3]Tsinghua University

## ABSTRACT

Large Language Models (LLMs) have exhibited remarkable reasoning capabilities and become the foundation of language technologies. Inspired by the great success of code data in training LLMs, we naturally wonder at which training stage introducing code data can really help LLMs reasoning. To this end, this paper systematically explores the impact of code data on LLMs at different stages. Concretely, we introduce the code data at the pre-training stage, instruction-tuning stage, and both of them, respectively. Then, the reasoning capability of LLMs is comprehensively and fairly evaluated via six reasoning tasks in five domains. We critically analyze the experimental results and provide conclusions with insights. First, pre-training LLMs with the mixture of code and text can significantly enhance LLMs' general reasoning capability almost without negative transfer on other tasks. Besides, at the instruction-tuning stage, code data endows LLMs the task-specific reasoning capability. Moreover, the dynamic mixing strategy of code and text data assists LLMs to learn reasoning capability step-by-step during training. These insights deepen the understanding of LLMs regarding reasoning ability for their application, such as scientific question answering, legal support, etc.

## 1 INTRODUCTION

Recently, Large Language Models (LLMs) have achieved impressive generalization performance across various tasks. Significantly, OpenAI developed ChatGPT OpenAI (2023a), Google designed PaLM Chowdhery et al. (2022), Baidu built ERNIE Bot Baidu (2023), and Alibaba presented Tongyi Qianwen Alibaba (2023). However, these industrial products are regrettably not open-source for commercial reasons. Thanks to the surging open-source projects of LLMs such as LLaMA (Touvron et al., 2023), Code LLama Roziere et al. (2023), Baichuan Yang et al. (2023), Alpaca (Taori et al., 2023), and ChatGLM (Du et al., 2022a), the academic research and industrial products of LLMs mark new milestones.

Two of the key factors to the great success of LLMs are 1) training data and 2) training strategies. First, for the training data, researchers aim to endow LLMs with language capabilities and general knowledge via training models on large-scale data from various domains. For example, LLaMA was trained with 1.4 trillion tokens consisting of texts (CommonCrawl, C4) and codes (GitHub). These large-scale data with diversity help the model to achieve competitive performance on multiple tasks. Second, the common pipeline goes through two stages for the training strategies: pre-training and instruction-tuning. The pre-training is conducted in a self-supervised manner on the massive unlabeled data, while instruction-tuning aims to fine-tune models with human-annotated prompts and feedback (Ouyang et al., 2022). Benefiting from the data and training strategies, LLMs gain remarkable skills, such as translation, conversation, examination, legal support, etc. These skills are all based on one of the most important capabilities, i.e., reasoning capability. So, how can LLMs gain such strong reasoning capability?

---

*Co-first author.
†Corresponding author.

We analyze the reasons from two aspects: training data and strategies. First, from the training data aspect, compared with the common textual data, code data is more logical and less ambiguous (refer to case studies in Appendix F). Also, from the experiments, researchers (Liang et al., 2022; Fu et al., 2022) verified that models trained on code data have strong reasoning capability. Therefore, code data is essential for model reasoning. Second, for the training strategies, both pre-training and fine-tuning are crucial to the model's performance. Pre-training feeds general knowledge to models while fine-tuning feeds domain-specific ability to models. To further explore the deep-in reasons for the strong reasoning capability of LLMs, this paper aims to answer an important question: at which stage does code data help LLMs reasoning?

To this end, we conduct comprehensive and fair experiments and provide analyses and conclusions with insights. First, we pre-train LLMs with pure text data and mixture data of code and text, respectively. Subsequently, at the instruction-tuning stage, LLMs are fine-tuned with the pure text data and mixture data of code and text, respectively. After training, to comprehensively measure the model reasoning capability, we evaluate LLMs on six tasks in five domains, including logical reasoning, code reasoning, legal reasoning, scientific reasoning, and analogical reasoning. Based on extensive experimental results and analyses, we provide three insights. 1) Pre-training LLMs with the mixture of code and text can significantly enhance LLMs' general reasoning capability almost without negative transfer on other tasks. 2) At the instruction-tuning stage, code data endows LLMs the task-specific reasoning capability. 3) The dynamic mixing strategy of code and text data assists LLMs to learn reasoning capability step-by-step during training. These findings deepen the understanding of LLMs regarding reasoning ability for their applications, such as scientific question answering, legal support, etc. The main contributions of this work are summarized as follows.

- Research question: this paper raises and aims to answer one essential concern, i.e., at which training stage can codes data help LLMs reasoning.

- Analyses and insights: we conduct extensive experiments and provide critical analyses and insights, which deepen the understanding of LLMs regarding reasoning capability.

- Open-source resource[1]: we release the model implementation and the trained model parameters, which contribute to the further research in the LLMs community.

## 1.1 TRAINING DATA & TRAINING STRATEGIES

Three key factors to the great success of LLMs are training data, training strategies, and model designs. In this section, we introduce our training data and training strategies. The next section details the model designs.

We study two training phases of LLMs, i.e., pre-training stage and instruction-tuning stage, on two different datasets including one plain text data and one text-code-mixed data. Figure 1 demonstrates the process of each stage. Specifically, we use the open-sourced PanGu2.6B and PanGu13B of the PanGu-$\alpha$ team Zeng et al. (2021) as baseline models for text models (trained on 100GB text data and larger text data, respectively), and train CodePanGu2.6B from scratch on the mixed code data for comparison. We will introduce detailed data settings in later chapters.

## 1.2 PRE-TRAINING CORPUS

The pre-training corpus consists of two parts. To ensure a fair comparison with PanGu2.6B, we collected a large amount of original data from public datasets such as BaiDuQA, CAIL2018, Sogou-CA, and network data sets such as Common Crawl, encyclopedias, news, and e-books according to the PanGu-$\alpha$ team (Zeng et al., 2021). Then we use rule-based data cleaning and model-based data filtering methods to filter to ensure high quality. Finally, we obtain 100GB of text data with the same scale and source as PanGu2.6B by sampling each data source using different ratios. Please refer to Appendix G for a detailed data processing process. To verify the influence of code data on the reasoning capability of the model in the pre-training stage, we used the CodeParrot (Huggingface, 2023) dataset as the second supplementary part. CodeParrot is a public Python dataset from BigQuery, comprising approximately 50GB of code and 5,361,373 files. Figure 2 shows the composition of the ~42B tokens in pre-training data.

---

[1] https://github.com/yingweima2022/CodeLLM

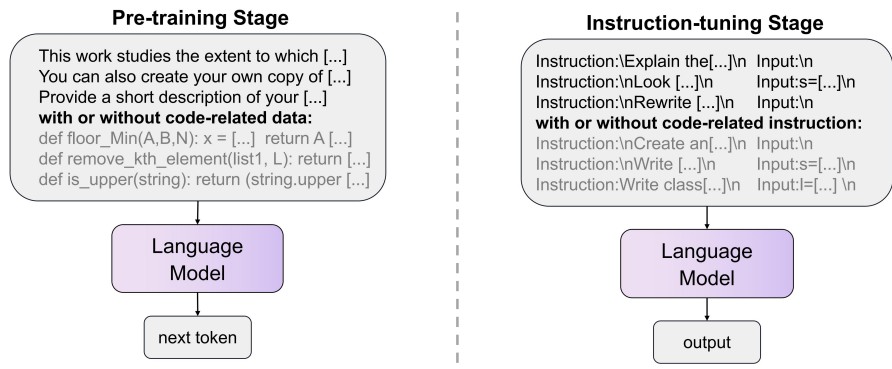

Figure 1: Demonstration of the pre-training and tuning phase.

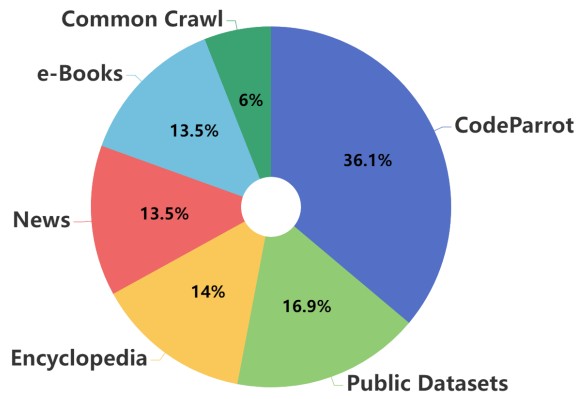

Figure 2: Distribution of the ∼42B tokens in pre-training data.

### 1.3 INSTRUCTION-TUNING CORPUS

We collect and construct 400K instruction tuning data to verify the effect of adding code instructions in the instruction tuning stage and convert them into a unified instruction format. The instruction tuning corpus is divided into two parts. The first part is from the natural language open source instruction dataset, Alpaca-GPT-4 (Peng et al., 2023) and PromptCLUE (pCLUE team, 2022). Alpaca-GPT-4 is generated by GPT-4, including 52K Chinese and English instruction tuning data. PromptCLUE unifies the differences between different NLP tasks (*e.g.*, reading comprehension, question answering) and converts the original task training set into a unified text-to-text data form, from which we randomly sample 200K data for instruction tuning.

The second part comes from the open-source data CodeAlpaca (Chaudhary, 2023) and our build dataset, with 150K instructions. The CodeAlpaca data contains 20K instruction tuning data generated according to the self-instruct technology, which can be used for instruction tuning of the code generation model. In order to supplement the code-related instruction tuning data, we use the CosQA (Huang et al., 2021) training set and the MBPP (Austin et al., 2021) training set to unify the task format in the way of PromptCLUE and expand the CodeAlpaca data. Figure 3 is an example of the format of instruction tuning data.

## 2 MODEL

We conduct experiments on large-scale autoregressive language models by adopting the GPT paradigm(Brown et al., 2020). It iteratively takes all tokens in the corpus as input, predicts the next token, and compares it to the ground truth. Assuming that a sequence $\mathcal{X} = \{x_1, x_2, ..., x_n\}$ is

Figure 3: Example of the instruction tuning data format. NL denotes natural language.

composed of $n$ tokens, the training objective can be formulated as maximization of the log-likelihood:

$$\mathcal{L} = \sum_{i=1}^{n} \log p(x_i | x_1, x_2, ..., x_{i-1}; \Theta) \tag{1}$$

where $p(x_i | x_1, x_2, ..., x_{i-1}; \Theta)$ is the probability of observing the $i$-th token $x_i$ given the previous context $x_1, x_2, ..., x_{i-1}$, and $\Theta$ denotes the model parameters.

## 2.1 MODEL ARCHITECTURE

Similar to recent pre-trained models such as GPT-3 Brown et al. (2020), LLaMA (Touvron et al., 2023), CodeGeeX (Zheng et al., 2023), and PANGU-$\alpha$ (Zeng et al., 2021), we follow a generative pre-training (GPT) architecture for autoregressive language modeling. At the same time, to make a fair comparison with the baseline of PanGu2.6B, we retain the setting of the 32-layer transformer decoder. The original GPT model uses a pooler function to obtain the final output. Follow CodeGeeX (Zheng et al., 2023) and PANGU-$\alpha$ (Zeng et al., 2021), we use an additional query layer on top of the stacked Transformer layers to explicitly induce the expected output with attention to obtain the final embedding.

## 2.2 TOKENIZATION

For the text-only model, we use the open-source vocabulary of the PanGu2.6B model released by PanGu-$\alpha$ team (Zeng et al., 2021), and the size of the vocabulary is 40,000. For the model training with mixed code, considering that there may be variables, functions, and class names in the code that are often meaningful words, we use the ChatGLM (Du et al., 2022b) vocabulary open-sourced by the THUGLM team to encode text and the code. The vocabulary size is 130,044. In addition, ChatGLM encodes multiple spaces as extra tokens to improve encoding efficiency. Specifically, L spaces are represented by <|extratoken_X|>, where X=8+L. Both vocabularies are BPE-based tokenizers, which use fixed-size vocabularies to handle variable-length characters in open-vocabulary problems.

## 3 EXPERIMENTS

## 3.1 TASK DESCRIPTION

To measure the reasoning ability of the models, we evaluate it on six tasks in realistic reason-centric scenarios, including general reasoning scenarios such as logical reasoning, legal reasoning, scientific reasoning, and analogical reasoning, and code-related scenarios such as code generation. These reasoning-intensive tasks elucidate the reasoning capabilities of the model through the model's performance in these scenarios. When publicly available, we evaluate the models with the test sets for each task. Otherwise, we use the development sets instead. We describe each task as follows.

**Logical Reasoning.** Logic is the study of reasoning and argumentation, which focuses on the rules of logic and methods of reasoning in the thinking process. We use the **logic** subject in the **C-Eval** dataset (Huang et al., 2023) to determine whether the model can understand and apply logical rules to make reasonable reasoning.

| Task Type | Dataset | Input & Prompt |
|---|---|---|
| Logical | Logic | The answer: $choice, can answer the following questions: $problem |
| Legal | JEC-QA | The answer: $choice, can answer the following questions: $problem |
| Scientific | ScienceQA | $lecture\n anwser: $choice can answer the following question: $question |
| Analogical | E-KAR | The reasoning relationship: $r1, the analogy reasoning relationship: $r2 |
| Code | CosQA | $question? Answered code is correct or wrong: $code |
| Code | MBPP | $question\n Code:\n |

Table 1: The input & prompt template for each task. $ is the input and other words are prompt.

**Legal Reasoning.** For legal reasoning, we use **JEC-QA** (Zhong et al., 2020), the largest question answering dataset in the legal domain, collected from the National Judicial Examination of China. The examination is a comprehensive evaluation of the professional skills of legal practitioners. Multiple reasoning skills are required to retrieve relevant material and answer legal questions.

**Scientific Reasoning.** We use the **ScienceQA** dataset (Lu et al., 2022) to evaluate the scientific reasoning ability of the model. The scientific question answering task can diagnose whether the artificial intelligence model has multi-step reasoning ability and interpretability. To answer scientific questions from ScienceQA, a model not only needs to understand multimodal content but also needs to extract external knowledge to arrive at the correct answer.

**Analogical Reasoning.** We use the **E-KAR** dataset Chen et al. (2022) to evaluate the model's analogical reasoning ability. It comes from the Civil Service Examination, a comprehensive test of the candidate's critical thinking and problem-solving ability. To solve the analogy reasoning problem, candidates need to understand the relationship among the options, which requires specific reasoning ability and background knowledge, especially common sense and facts, and knowing why a fact is denied.

**Code Reasoning.** We use **CosQA** (Huang et al., 2021) to test the model performance on the code question-answering task. The dataset includes 604 natural language-code question-answer pairs. Furthermore, we use the **MBPP** dataset (Austin et al., 2021) to test the model code generation ability, containing 427 Python coding questions.

## 3.2 EVALUATION DETAILS

In evaluation, these tasks are usually divided into two parts, understanding task and generation task. For the understanding task, we follow PanGu2.6B (Zeng et al., 2021) and CPM Zhang et al. (2021), decomposing the task into a perplexity comparison task. We construct a prompt template for each evaluation task and populate the template with instances as input to the model. Table 1 describes the templates for each task.

We adopt a perplexity-based approach to solve classification tasks. For each <text, label> pair, input will be automatically generated according to the predesigned prompt in Table 1. The sequences generated by the prompt will be fed into the model, and a perplexity value will be calculated. The label corresponding to the minimum perplexity value will be regarded as the predicted label for this passage. For the generative task, we leverage the properties of autoregressive language models to generate corresponding answers directly from a given input naturally.

## 3.3 RESULTS

### 3.3.1 PRE-TRAINING STAGE

To illustrate the impact of code data in the pre-training phase on the reasoning capabilities of large language models, we compared the performance of the three models in real reasoning-intensive scenarios. Among them, the NL (2.6B) and NL (13B) (*i.e.*, PanGu2.6B and PanGu13B) models (Zeng et al., 2021) are trained on natural language datasets, and the CODE (2.6B) (*i.e.*, CodePangu2.6B) model is trained on mixed data (the dataset mentioned in Chapter 1.2). The models are evaluated in zero-shot manner on downstream tasks. Specifically, we report accuracy on for Logic, JEC-QA, ScienceQA, E-KAR, and CosQA tasks and BLEU score for MBPP task. In order to illustrate the

| Dataset | Task | Metric | NL (2.6B) | NL (13B) | CODE (2.6B) | p-value |
|---------|------|--------|-----------|----------|-------------|---------|
| Logic* | Logical Reasoning | ACC | 36.36 | **45.45** | 40.90 | 4.197e-06 |
| JEC-QA* | Legal QA | ACC | 27.00 | 27.00 | **28.70** | 1.956e-25 |
| ScienceQA* | Scientific QA | ACC | 45.93 | 45.18 | **46.06** | 0.014 |
| E-KAR* | Analogical Reasoning | ACC | 32.24 | 35.52 | **36.12** | 7.013e-07 |
| CosQA† | Code QA | ACC | 47.02 | 46.85 | **50.50** | 1.066e-40 |
| MBPP† | Code Generation | BLEU | 0.52 | 1.34 | **5.06** | - |

Table 2: Results on pre-training stage. Bold values indicate the best performance. * denote the general reasoning task, and † denote the code-related reasoning task.

significance of these results, we conducted a t-test on the predicted scores. It demonstrates that all p-values are less than 0.05, indicating that the results are statistically significant. Table 2 depicts the results of these tasks. Consistently over these tasks, we have two observations as follows.

- After adding code training, LLM performs better on most reasoning-related tasks, even though most of these tasks are not related to code. This shows that adding code data in the pre-training stage can not only improve the coding-related ability but also improve the general language reasoning ability of the model to a certain extent.

- Even with a larger scale model, *i.e.*, NL (13B), it is still not as effective as CODE (2.6B) in these reasoning scenarios. This is similar to the results of HELM Liang et al. (2022), which suggest that if (a) the computational budget is constrained and (b) the resulting model is applied in the code reasoning domain, adding code data in the pre-training phase may be more effective than increasing the model parameter size.

In summary, we find that simply adding code data during the pre-training phase can effectively improve the model's general reasoning ability, which might indicate that mixing more code data for training may produce a competitive model to solve tasks that require complex reasoning to complete. This provides a promising prospect for subsequent LLM development.

### 3.3.2 INSTRUCTION-TUNING STAGE

ChatGPT OpenAI (2023a) and GPT-4 OpenAI (2023b) successfully use instruction tuning to enable LLMs to follow natural language instructions and complete real-world tasks; this improvement has become standard in open-source LLMs. This is implemented by fine-tuning the model on a wide range of tasks using human-annotated instructions and feedback, by supervised fine-tuning via manually or automatically generated instructions using public benchmarks and datasets, or learning from instruction-following data by developing from state-of-the-art instruction-tuned teacher LLMs.

To illustrate the impact of code data on the LLMs reasoning ability in the instruction tuning stage, we use the instruction tuning datasets that contain codes and the instruction tuning datasets without codes introduced in Chapter 1.3 to fine-tune the PanGu2.6B and PanGu13B models (Zeng et al., 2021) and evaluate their performance in reasoning-intensive scenarios. In addition, we also fine-tune the CodePanGu2.6B model using the instruction tuning dataset containing codes to observe the effect of using code data in both pre-training and instruction tuning stages. Table 3 shows the results of these tasks. Among them, NN and NC represent the fine-tuned PanGu model using only text instructions and instructions containing codes, respectively, and CC represents the fine-tuning model of CodePanGu2.6B using instructions containing codes. Consistently over these tasks, we observe the following:

- After fine-tuning with mixed code instruction data, LLM shows different trends in multiple reasoning tasks. This indicates that introducing code data in the instruction tuning phase may be less effective than in the pre-training phase. Therefore, it is best to add code data in the pre-training stage to improve the model performance in general reasoning tasks.

- We find that training with code data in both stages can significantly improve code-related tasks (CosQA and MBPP), especially code generation tasks. This may be because the code instruction data activates the code reasoning ability of the language model, which suggests that if the LLM needs to complete complex code tasks, the code reasoning ability can be improved by effectively following code instructions and generating compliant content.

| Dataset | NN (2.6B) | NC (2.6B) | NN (13B) | NC (13B) | CC (2.6B) |
|---|---|---|---|---|---|
| Logic* | 36.36 | **40.90** | **40.90** | **40.90** | **40.90** |
| JEC-QA* | 25.20 | 26.10 | 24.50 | 26.40 | **27.10** |
| ScienceQA* | **44.45** | 43.44 | 42.94 | 43.41 | 41.90 |
| E-KAR* | **30.45** | 28.66 | 26.27 | 27.46 | 27.20 |
| CosQA† | 45.20 | 48.18 | 47.52 | 51.99 | **52.48** |
| MBPP† | 0.00 | 5.61 | 0.00 | 1.88 | **24.88** |

Table 3: Results on instruction-tuning stage. Bold values indicate the best performance. * denote the general reasoning task, and † denote the code-related reasoning task.

| Dataset | NL (2.6B) | CODE (2.6B) | NL (2.6B)+CoT | CODE (2.6B)+CoT |
|---|---|---|---|---|
| ScienceQA | 45.93 | 46.06 | 68.76 | **70.30** |
| E-KAR | 32.24 | 36.12 | 69.55 | **72.84** |

Table 4: Results with Chain-of-Thought prompt. Bold values denote the best results.

- Compared with the pre-training stage, the performance of instruction-tuned LLMs on some tasks is degraded, similar to the TÜLU (Wang et al., 2023) results. This may be because the instruction tuning data usually covers a wide range of domains and dialogue content, causing the model to tend to answer questions more comprehensively, resulting in a decline in reasoning ability. We propose that if specific reasoning capabilities are required, they can be augmented by adding domain-specific instructions during the tuning phase.

In summary, we find that adding code data in the instruction tuning stage is not as effective as the pre-training stage in improving the general reasoning ability of the model. However, we find that code instructions made the model follow natural language instructions and generate correct code, improving the model's code reasoning ability. This also suggests that tuning with relevant data may be helpful when solving specific reasoning tasks.

### 3.3.3 CHAIN-OF-THOUGHT ABILITY

Compared with the standard prompt technology, Chain-of-Thought (CoT) (Wei et al., 2022) transforms tasks into a continuous chain generation process. This technology enhances the model ability in complex reasoning tasks by providing a language model with a series of related reasoning steps. To evaluate the potential of the model in utilizing chains of thought in solving complex problems, we conduct experiments on two pre-trained models, NL (2.6B), *i.e.*, PanGu2.6B and CODE (2.6B), *i.e.*, CodePanGu2.6B on ScienceQA(CoT) (Lu et al., 2022) and E-KAR(CoT) (Chen et al., 2022) datasets. We incorporate CoT information as a part of the model input with the question and context information. In this way, the model can directly use the reasoning process of the thinking chain for answer generation. The experimental results are shown in Table 4.

The experimental results show that after the introduction of the Chain-of-Thought, the performance of all models in reasoning problems is significantly improved by making full use of the coherent reasoning process of CoT. The CoT information is used as part of the model input to help the model better understand the problem and generate answers according to the logic of the CoT. Among them, CODE (2.6B) achieved the best performance, indicating that CODE (2.6B) can better use CoT information for reasoning. This also suggests that pre-training with mixed-code data may result in a competitive model for tasks that require complex reasoning.

### 3.3.4 EXPLORING WAYS TO MIX CODE AND TEXT DATA

Previous experiments have demonstrated that training with mixed code data in the two stages of pre-training and instruction tuning can improve the general and specific reasoning capabilities of

LLMs, respectively. Therefore, We naturally wonder how mixing these two types of data can better improve model reasoning ability, which has not been explored in previous studies. Therefore, we design comparative experiments in the instruction tuning stage to verify the impact of different data mixing strategies. The mixed strategy is shown in Table 5. One group is uniform sampling, that is, the proportion of text and code in each group of training data is roughly the same; the other two groups gradually increase or decrease the proportion of code to verify whether step-by-step learning will better activate the reasoning ability of LLMs. The experimental results are shown in Table 6.

| Phase | Uniform Sampling | Stepwise Increase | Stepwise Decrease |
|---|---|---|---|
| 1 | 5:3 | 7:3 | 5:5 |
| 2 | 5:3 | 7:3 | 6:4 |
| 3 | 5:3 | 6:4 | 7:3 |
| 4 | 5:3 | 5:5 | 7:3 |

Table 5: Mixture strategies on text data and code data with different ratios (text:code).

| Dataset | Uniform Sampling | Stepwise Increase | Stepwise Decrease |
|---|---|---|---|
| Logic* | 31.82 | 36.36 | **40.90** |
| JEC-QA* | **27.30** | 26.70 | 27.10 |
| ScienceQA* | **43.76** | 43.19 | 41.90 |
| E-KAR* | **28.66** | 28.36 | 27.46 |
| CosQA† | 51.65 | 50.66 | **52.48** |
| MBPP† | 23.68 | 23.42 | **24.88** |

Table 6: Result of different mixed strategies. Bold values indicate the best performance. * denote the general reasoning task, and † denote the code-related reasoning task.

The experiment showed that the training strategy of using a higher code data ratio in the early stage and gradually reducing the code data ratio in the later stage achieved the best results in code question answering (CosQA) and code generation (MBPP) tasks, while ensuring the performance of the model in other reasoning tasks. This may be because, due to the strong logic of the code, using more code data in the early stage may help the model activate the code reasoning ability faster. Therefore, if LLMs are expected to have better specific reasoning ability, adopting a stepwise descent strategy can better activate the model potential. In addition, since experiments in the pre-training phase require a lot of resources, we leave the validation of this phase to later work.

### 3.3.5 OTHER TASKS

We extensively evaluates various reasoning tasks, including logical and code reasoning, highlighting the positive impact of code-related data. Additionally, we sought to ascertain whether code data would affect common-sense tasks. Therefore, to verify the impact of code data on other comprehension and generation tasks that are less demanding on reasoning, we conduct experiments on other tasks, including two NLI tasks (OCNLI (Hu et al., 2020) and CMNLI (Wang et al., 2018)), requiring the model to identify the relationship between two sentences, either entailment, neutral or contradiction; a free-form multiple-choice Chinese machine reading comprehension dataset ($C^3$) (Sun et al., 2020)

| Dataset | Metrics | without code | with code |
|---|---|---|---|
| $C^3$ | ACC | 54.14 | **54.30** |
| OCNLI | ACC | **41.69** | 40.50 |
| CMNLI | ACC | **45.07** | 43.49 |
| DuReader | EM | **0.42** | 0.14 |
| | F1 | **15.29** | 8.73 |

Table 7: Results of pre-training on other tasks. Bold values indicate the best performance.

| Dataset | Metrics | without code | with code |
|---------|---------|--------------|-----------|
| $C^3$ | ACC | **55.07** | 54.47 |
| OCNLI | ACC | 40.78 | **41.19** |
| CMNLI | ACC | 44.82 | **45.49** |
| DuReader | EM | **12.07** | 8.05 |
| | F1 | **34.85** | 25.05 |

Table 8: Results of instruction-tuning on other tasks. Bold values indicate the best performance.

consisting of documents (conversational or more formal mixed-type text) and their associated multiple-choice free-form questions; one reading comprehension task duReader (He et al., 2017), requiring the model to extract a text span from a given paragraph as the correct answer to the question. Refer to Appendix D for prompt templates and evaluation metrics for different tasks.

Table 7 and Table 8 show the results of adding code data in the pre-training phase and adding code instructions in the instruction tuning phase (only in this phase). Experimental results show that, in most cases, using code data for training may negatively impact the performance of other tasks. In the DuReader reading comprehension task, part of the performance will be reduced after adding code at different stages. This may be because the model does not thoroughly learn the code and text data, resulting in confusion when the model generates answers to reading comprehension questions. In future, we will verify and solve it in a larger model and with larger data.

## 4 RELATED WORK

**LLM training.** LLMs are usually based on the transformer architecture (Dai et al., 2019). Notable models include BERT (Devlin et al., 2018), GPT-2 (Radford et al., 2019), and T5 (Raffel et al., 2020); after the emergence of GPT-3 (Brown et al., 2020) with 175B parameters, a batch of larger models emerged, including PaLM (Chowdhery et al., 2022), OPT (Zhang et al., 2022), PanGu-$\alpha$ (Zeng et al., 2021), and LLaMA (Touvron et al., 2023), which have achieved remarkable results on various NLP tasks. For LLMs to follow instruction output, instruction tuning (Peng et al., 2023) plays an important role. This can use human-annotated feedback (Ouyang et al., 2022) or public benchmarks to automatically generate instructions (Wang et al., 2022) to fine-tune models on various tasks.

**Model Evaluation.** HELM Liang et al. (2022) provides a large-scale evaluation of existing LLMs. They found that the code-cushman-001 model trained on code data had more reasoning power than other natural language models. In addition, some work analyzed the reasoning ability of LLMs through the way of CoT and found that because the original GPT-3 had not been subjected the code training, it could not do CoT reasoning Wei et al. (2022). The PaLM training data contains 5% of the code training data, which shows that PaLM can effectively perform CoT reasoning. However, in the above work, the evaluated models have different parameters, data sizes, and unknown training details. Therefore, it is not rigorous to speculate the exact impact of code data on reasoning ability only by comparing existing models.

## 5 CONCLUSION

In this paper, we investigate at which stage introducing code data can help improve the reasoning ability of LLMs. We validate the effect of code data at different stages with the same parameter scale and using the same training objective. We point out that simply adding code data in the pre-training phase can effectively improve the general reasoning ability of the model. Furthermore, we find that adding code instructions in the instruction tuning stage can make the model follow human instructions for output and improve specific code reasoning capabilities. Moreover, we point out that the dynamic mixing strategy of code and text data assists LLMs in learning reasoning capability step-by-step during the training process. We provide a well-designed and tested reference implementation for LLMs training to help researchers and developers better understand and analyze LLMs. Models such as ChatGPT illustrate that larger-scale models will produce emergent capabilities. Therefore, the follow-up of this paper will study how to code the impact of data on models of different sizes and explore the relationship between code data and emergent capabilities.

ACKNOWLEDGMENTS

This research was funded by NSFC No.62272473, the Science and Technology Innovation Program of Hunan Province (No.2023RC1001) and NSFC No.62202474.

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

## A    ADDITIONAL RELATED WORK

**Data Mixtures.** Models such as GPT-3 (Brown et al., 2020) and PanGu-$\alpha$ (Zeng et al., 2021) are trained on natural language data from various domains, and models such as LaMDA (Thoppilan et al., 2022) and LLaMA (Touvron et al., 2023) are additionally trained on code data. However, the impact and specific origin of this mixed-code data is unclear. Some researchers have extensively analyzed the performance of current LLM on various tasks, pointing out that code may be the key to improving reasoning ability (Liang et al., 2022; Fu et al., 2022). However, the evaluated models have different parameters and data scales, and problems such as unknown training details exist. It is difficult to determine the exact impact of code data on the reasoning ability of LLMs.

# B    CODE REPRODUCTION

## B.1    MINDSPORE VERSION

To ensure reproducibility, we open-sourced the model training and inference code using the Mind-Spore framework in the repository link: `https://github.com/yingweima2022/CodeLLM`.

## B.2    INTRODUCTION TO MINDSPORE

We acknowledge that MindSpore is relatively newer and requires further development time. However, we would also like to highlight the adoption of MindSpore by prominent researchers in the field. For instance, (Zheng et al., 2023) in their work on CodeGeeX, (Liu et al., 2021)'s OPT model and (Christopoulou et al., 2022) in the release of PanGuCoder have both embraced the software-hardware combination offered by MindSpore. According to the data from Papers with Code (`https://paperswithcode.com/trends`), there have been 398 repositories utilizing the open-source MindSpore framework since 2023, which is higher than the 192 repositories using TensorFlow. This showcases its potential and the growing interest within the community.

# C    EXPERIMENTS DETAILS

Our experiments are developed under the Mindspore framework. To make a fair comparison with the baseline of PanGu2.6B, we retain the setting of the 32-layer transformer decoder. The model architecture as shown in Figure 4. In the pre-training stage, we trained CodePanGu2.6B on a cluster of 16 Ascend 910 AI processors, and in the instruction-tuning stage, we tuned models on a cluster of 8 Ascend 910 AI processors. The sequence length for the training data is set to 1024 for all the models. Other detailed configurations can be found in Table 9.

| Type | Parameter | Value |
|---|---|---|
| Environmental parameter | Framework | Mindspore v1.7.0 |
| | Hardwares | Ascend 910 |
| | Mem per GPU | 32GB |
| | GPUs per node | 8 |
| Model parameter | Layers | 32 |
| | Hidden size | 2560 |
| | FFN size | 10240 |
| | Heads | 32 |
| Optimization parameter | Optimizer | Adam |
| | Initial/final learning rate | 1e-4(2e-5)/1e-6 |
| | Warm-up step | 500 |
| | Learning rate scheduler | cosine |
| | Optimizer parameters | $\beta1 = 0.9, \beta2 = 0.95$ |
| Parallelism parameter | Data parallel | 16(8) |
| | Model parallel | 1 |
| | pipeline parallel | 1 |

Table 9: Training configurations.(The values in parentheses are instruction-tuning parameters)

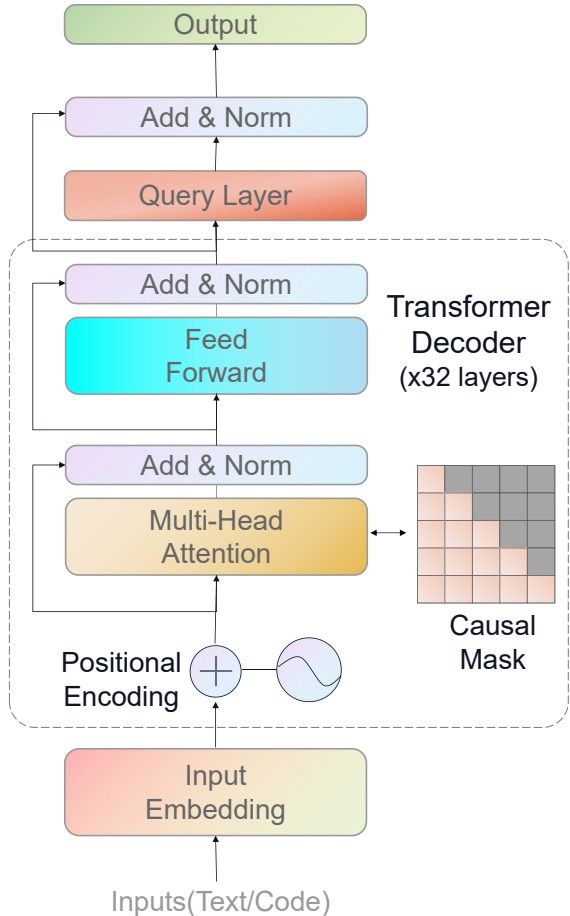

Figure 4: Model architecture. We build models with 2.6B and 13B parameters, consisting of 32-layer left-to-right transformer decoders and a top query layer.

## D  THE TEMPLATE FOR OTHER TASKS

We follow Chapter 3.2, conduct experiments on other tasks to verify the impact of code data on other comprehension and generation tasks that are less demanding on reasoning, including $C^3$ (Sun et al., 2020); two NLI tasks (OCNLI (Hu et al., 2020) and CMNLI (Wang et al., 2018)); one reading comprehension task duReader (He et al., 2017). Table 10 shows the prompt templates for these tasks. The evaluation metrics for duReader, including F1 and exact match(EM), measure the similarity between the predicted and ground-truth text spans. The evaluation metric of other tasks is accuracy.

| Task | Input & Prompt |
| --- | --- |
| $C^3$ | Question: $question\n Answer:$choice comes from the dialogue: $context |
| OCNLI | $S1? Yes/Maybe/No, $S2 |
| CMNLI | $S1? Yes/Maybe/No, $S2 |
| duReader | Read document: $Document\n Question:$Question \n Answer: |

Table 10: The input & prompt template for other tasks.

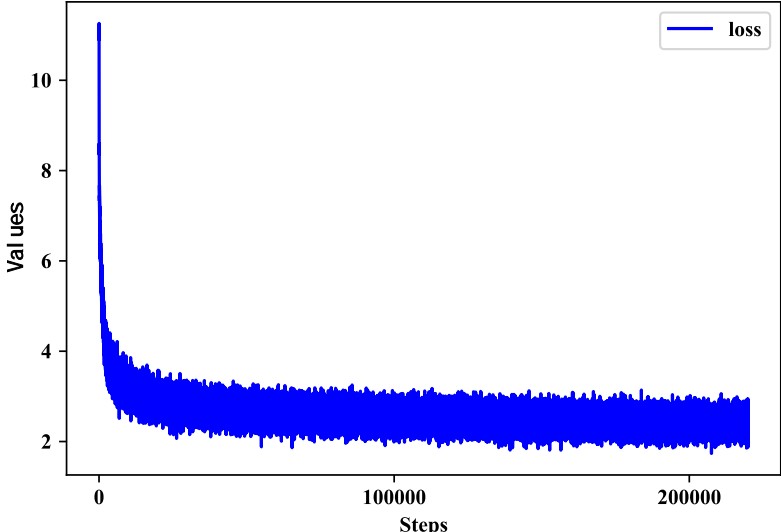

Figure 5: The curves of training loss for CodePanGu2.6B.

## E  TRAINING LOSS

The curves of training loss for the CodePanGu2.6B model are shown in Figure 5. We show that the cross entropy loss decreases steadily during training and the loss of this model converges to around 2.25.

## F  CASE STUDY

In summary, adding code data in the pre-training stage can effectively improve the general reasoning ability of LLM, and can guide the model to make full use of the coherent reasoning process of the Chain-of-Thought to generate answers. Consistent with GPTRoadMap's point of view (Fu et al., 2022), we think this may have something to do with the logic of the code itself. To further explain why the code improves the reasoning ability of the model, we found several sample codes from the dataset and explained each code, as shown in Figure 6.

We found that, regardless of the length of the code dealing with different problems, step-by-step reasoning is required to ensure that the code is generated correctly, similar to the Chain-of-Thought required by other reasoning tasks. This may indicate that the model implicitly learns the thinking chain ability through the code data, which improves the reasoning ability of the language model. In addition, we analyzed the data flow graph of the *calculate_average* function, as shown in Figure 7. We found many data flow dependence relations in the code data, which are distributed among different code variables. Complex reasoning tasks usually require long dependencies to infer correct conclusions, so the language model may benefit from dependencies such as data and control flow of code data and improve the reasoning ability of the model.

## G  DATASET CONSTRUCTION

**Cleaning and Filtering.** To improve the data quality, we adopt the following rule-based text cleaning strategies over the raw web pages from Common Crawl.

- Remove the document which contains less than 60% Chinese characters.
- Remove the document which contains less than 150 characters.
- Remove the document which contains only the title of a webpage.
- Remove the special symbols and duplicated paragraphs in each document.

| Description | Code | Explanation |
|---|---|---|
| Write a funtion to implement quick sort | ```python
def quicksort(arr):
    if len(arr) <= 1:
        return arr
    pivot = arr[0]
    less = [x for x in arr[1:] if x <= pivot]
    greater = [x for x in arr[1:] if x > pivot]
    return quicksort(less) +
        [pivot] + quicksort(greater)
``` | 1.Choose the first element as the pivot

2.Create a list of elements smaller than or equal to the pivot

3.Create a list of elements greater than the pivot

4.Recursively sort the list |
| Write an online shopping system based on python | ```python
class Book:
    def __init__(self, title, category):
        self.title = title
        ...
    def borrow(self, borrower):
        ...
class Library:
    def __init__(self):
        self.books = []
        self.borrow_history = []
        def borrow_book(self, title,
                        borrower):
        ...
``` | 1.Create a book class (Book):
**Attributes:**
title, author, classification, ...
**method:**
borrow, return_book, Display book information (display_info), ...

2.Create a library class (Library):
**Attributes:**
Book list (books), Borrowing History (borrow_history), ...
**method:**
Add Book (add_book), Remove Book (remove_book), ... |

Figure 6: Examples of different codes.

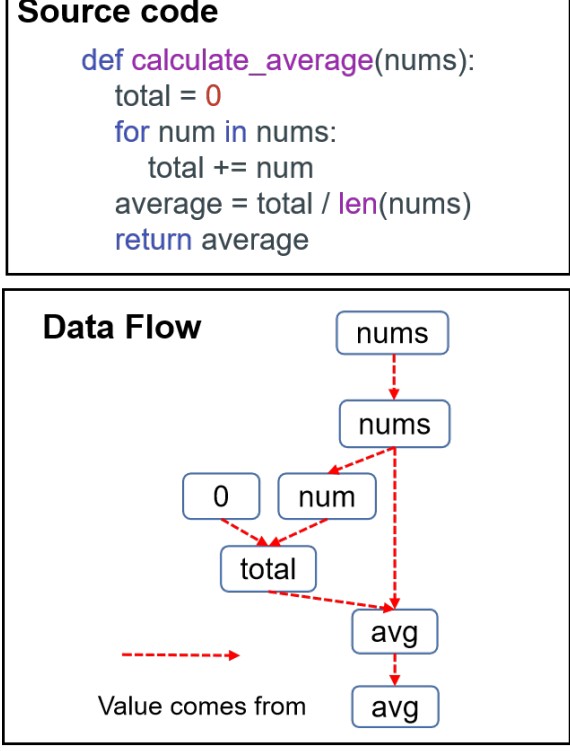

Figure 7: Examples of code dependencies.

- Identify advertisements based on keywords and remove documents that contain advertisements.

- Identify the navigation bar of the web page and remove it.

Regarding the use of The Common Crawl corpus `https://commoncrawl.org/the-data/` in our work, it's worth noting that several prominent projects, such as Llama, have also employed this dataset. Crawled data may exhibit biases, and ground truth data often resides within large corporations, making access challenging. In light of this, we adopted a strategy similar to other open-source models like Llama and Falcon, leveraging a broader range of data types such as open-source code and e-books to supplement and mitigate potential biases.

**Text Deduplication.** Although we removed duplicate paragraphs in each document in the previous step, there are still documents with highly overlapping content in different data sources. Therefore, we carry out fuzzy data deduplication over the documents across all our data sources to further remove high-overlap content. For fuzzy data deduplication, we employed Spark's MinHashLSH algorithm, a widely adopted technique by models like GPT-3.

**Data Selection.** Using the construction process described above, we constructed filtered text corpora from five types of data sources. Based on this corpus, we constructed a training dataset of 100GB text data by sampling each data source according to the ratio of Figure 2 and used this data as the first part of the training set to train CodePanGu2.6B.

## H  URLs of Used Datasets

This section gives the URLs of the used benchmark datasets.

- The Common Crawl corpus: https://commoncrawl.org/the-data/
- BAAI-WuDao: https://openi.pcl.ac.cn/BAAI/WuDao-Data
- CodeParrot: https://huggingface.co/codeparrot/codeparrot
- github-code: https://huggingface.co/datasets/codeparrot/github-code
- stanford_alpaca: https://github.com/tatsu-lab/stanford_alpaca
- code_alpaca: https://github.com/sahil280114/codealpaca
- PromptCLUE: https://github.com/clue-ai/PromptCLUE

## I  Examples of Datasets

Below are examples from the Logic, JEC-QA, and E-KAR datasets.

**One Example of Logic (Logical Reasoning).**

Problem:

Regarding the physical education standard test for Class A, three teachers made the following predictions: Teacher Zhang said, "Not everyone will fail." Teacher Li said, "Someone will fail." Teacher Wang said, "Both the class president and the study committee member will pass." If only one of these teachers' predictions is correct, which of the following must be true?

Answer List:

A: "Both the class president and the study committee member failed." B: "Both the class president and the study committee member passed." C: "The class president passed, but the study committee member failed." D: "The class president failed, but the study committee member passed."

Answer: A

**One Example of JEC-QA (Legal QA).**

Problem:

A miscellaneous article written by person A caused a significant stir after its publication. The article was reprinted by several newspapers and websites without compensation. Person B translated the article into French and person C translated it into Uighur. Both translations were published domestically without A's consent and without any remuneration. Which of the following viewpoints is correct?

Answer List:

A: "The act of newspapers and websites reprinting the article does not constitute infringement." B: "The actions of both B and C do not constitute infringement." C: "B's action does not constitute infringement, but C's action does." D: "B's action constitutes infringement, but C's action does not."

Answer: C

**One Example of E-KAR (Analogical Reasoning)**

Problem:

Based on the given relationship [Speed:Time:Distance], choose the option that fits this relationship.

Answer List:

A: "Interest Rate:Principal:Interest" B: "Quality:Variety:Quantity" C: "Profit:Cost:Value" D: "Income:Expenditure:Surplus"

Answer: A

Explanation: From the given relationship, we infer the following: "Speed" multiplied by "Time" equals "Distance". In option A, "Interest Rate" multiplied by "Principal" equals "Interest". In option B, there is no clear logic connecting "Quality", "Variety", and "Quantity". In option C, the product of "Profit" and "Cost" is not "Value". In option D, "Surplus" is the difference between "Income" and "Expenditure". Therefore, the correct choice is A.

## J    ADDITIONAL CONCLUSIONS OF MIXTURE STRATEGY

From the extensive experiments of the mixture strategy, we have three additional conclusions as follows.

1) Using a larger code proportion in the early stage can improve the performance of LLM in coding tasks (CosQA and MBPP). The reason may be that a higher code proportion in the early stage can better activate code-related reasoning capabilities under a higher learning rate.

2) The descending strategy can improve the performance of logic. Since the code data is more logical, giving more codes in the initial stage may improve the performance of logic reasoning.

3) In the other three datasets, uniform is better, probably because these tasks require both logical reasoning ability and common sense as well as natural language understanding ability.

Therefore, we recommend choosing different hybrid strategies based on the characteristics of different downstream tasks.

## K    OTHER DATASETS

In order to more comprehensively verify the observations of this article, we selected the high school mathematics and high school physics problem parts of the MMLU (Hendrycks et al., 2020) test set to evaluate the model in the pre-training stage. MMLU is a currently widely used data set to evaluate the comprehensive ability of LLM (OpenAI, 2023b; Touvron et al., 2023), among which mathematics and physics can better reflect the reasoning ability of the model. The results obtained are shown in the table 11 below.

| Task | NL2.6B | NL13B | **CODE2.6B** | p_value | *LLaMA-7B* |
|------|--------|-------|--------------|---------|------------|
| MMLU_Math | 24.16 | 22.30 | **24.91** | <0.05 | *24.97* |
| MMLU_Physics | 20.00 | 22.67 | **26.67** | <0.05 | *27.97* |

Table 11: Performance in mathematics and physics subjects.

We can make the following observations:

- On mathematical and physical reasoning tasks, the CODE2.6B model shows advantages over NL2.6B and NL13B, which strengthens the effectiveness of introducing code data

in the pre-training stage. We performed a t-test for statistical significance and the results showed that the p-value was less than 0.05, which indicated that the results were statistically significant.

- We admit that models including LLaMA-7B and CODE2.6B perform relatively poorly on mathematical and physical tasks, but we believe that the relative improvement brought by code data is trustworthy. This also shows how it is necessary to further improve mathematics-related tasks.

Note: Source of LLaMA results: https://github.com/baichuan-inc/Baichuan-7B

In addition, we supplemented our pre-training with an additional 50GB of natural language data on top of the NL (2.6B) dataset. From Table 12, we observed performance improvements on some natural language datasets; however, these benefits were not as significant as those seen with code data. Moreover, there was no enhancement in performance on code datasets, and in some cases, there was even a negative effect. Therefore, it would be valuable to further explore the hybrid training of code and text data or even other types of modal data.

| Dataset | Logic | JEC-QA | ScienceQA | E-KAR | CosQA | MBPP |
|---------|-------|--------|-----------|-------|-------|------|
| NL(2.6B)+50G | 36.36 | 27.20 | 45.96 | 32.54 | 46.69 | 0.45 |

Table 12: Performance of model pre-trained with an additional 50GB data.

## L LIMITATIONS

### L.1 VERIFIED ON MORE LARGE LANGUAGE MODELS.

PanGu (Zeng et al., 2021), along with other LLMs like Llama (Touvron et al., 2023) and Google PaLM (Chowdhery et al., 2022), shares a common architecture based on the GPT-2 OpenAI (2023a) decoder-only architecture and next token prediction task. Due to resource constraints and environmental concerns, we currently only conduct experiments on PanGu, but we compare models with different parameter scales (*i.e.*, 2.6B and 13B) to demonstrate the impact of the code. For a more comprehensive validation on a larger model, this section analyzes the performance of two other models, namely the Llama 2 and Code Llama models on reasoning tasks. Among them, Code Llama continued to pre-train using code data based on Llama 2. The performance summary on code tasks and mathematical tasks is shown in Table 13. The data comes from Open LLM Leaderboard( `https://huggingface.co/spaces/HuggingFaceH4/open_llm_leaderboard`). Experimental results show that pre-training on code data can improve coding and mathematical reasoning capabilities, and also preliminarily verify some of the conclusions obtained in this article. In the future, this article will continue to verify and discuss model experiments on a larger scale. At the same time, we look forward to the emergence of LLMs with different architectures than GPT, and we also look forward to following up and verifying more LLMs with different architectures.

| Model | HumanEval (CODE) | GSM8K (MATH) |
|-------|------------------|--------------|
| Llama 2-7B | 12.2 | 3.49 |
| Code Llama-7B | **33.5** | **5.16** |
| Llama 2-13B | 20.1 | 10.84 |
| Code Llama-13B | **36.0** | **12.13** |

Table 13: Results on Llama 2 and Code Llama (7B and 13B).

### L.2 WHY NOT DESIGN A NEW MODEL FOR CODE?

Our paper primarily investigates at which training stage can codes help general LLMs reasoning. Through extensive experimentation, we assess the impact of code data, offering insights that can guide the development of future universal LLMs. Our objective is not to create a specialized code or text model, but rather to provide guidance in this context. In the future, we plan to leverage these findings for future development. We will explore integrating code features with models to create more robust and versatile universal reasoning models.

# M  APPLICATION

The main contribution of this paper is to explore the impact of code data on LLM reasoning capabilities at different training stages and draw the following conclusions. Firstly, adding code data in the pre-training stage can effectively improve the model's general reasoning capabilities. Secondly, adding code instructions in the instruction fine-tuning stage can improve specific code reasoning capabilities. Moreover, the dynamic mixing strategy of code and text data assists LLMs in learning reasoning capability step-by-step during training. Below, we provide some ideas for further applications of these conclusions.

## M.1  IMPROVE GENERAL REASONING SKILLS.

The conclusions obtained in this article have direct implications for the application of general LLMs in multiple reasoning-intensive fields (such as legal support, scientific question answering, etc.). Secondly, for the construction of large models in vertical fields (such as large legal models, etc.), it is often necessary to use vertical field data for continued pre-training. High-quality data is the key to model training. If there is a lack of data in vertical fields, you can try to mix certain high-quality code data to improve the model's reasoning capabilities in vertical fields.

## M.2  IMPROVE DOMAIN-SPECIFIC REASONING SKILLS.

This paper finds that mixing in code data during the fine-tuning phase can improve capabilities in specific areas of the code. This may be because instructions for related tasks may activate reasoning and response abilities in the corresponding tasks. Therefore, for specific fields such as information extraction Li et al. (2023), you can also consider mixing in instructions from specific fields to further improve reasoning capabilities.

## M.3  IMPORTANCE OF TRAINING DATA.

This paper verifies the impact of code data through ablation experiments and proves the importance of data. However, there are still data that may be similar to code data, such as scientific paper data, mathematical data, etc. This may also be the key to improving the reasoning ability of language models. The experimental exploration in this article provides a direction for further research on how multiple types of data affect model training.

