# OpenReview forum: "At Which Training Stage Does Code Data Help LLMs Reasoning?"
_ICLR.cc/2024/Conference — ICLR 2024 spotlight_

### Official Review · Reviewer_L6FP · 2023-10-28

**Soundness:** 3 good
**Presentation:** 3 good
**Contribution:** 3 good
**Rating:** 8
**Confidence:** 4

**Summary:**

In this paper, the authors aim to explore the impact of introducing code at different training stages of large language models (LLMs) and how it affects LLMs’ reasoning capability. The experiments are conducted by introducing code data in the pre-training stage, the instruction tuning stage, and both stages to evaluate LLM through six inference tasks in five domains. They provide deep-in insights via comprehensive experiments and critical analyses. The resources of this paper are released.

**Strengths:**

- This paper is well-motivated. The impact of code data in LLMs is a hot research question. This paper answers this issue from the reasoning capability aspect.

- The experiments are comprehensive, and the insights are remarkable. The reasoning capability of LLMs is evaluated via six tasks in five domains. The authors provide critical analyses and significant insights on training LLMs and the reasoning capability of LLMs.

- The idea of dynamic mixed strategy is easy to follow yet effective. It helps LLMs learn reasoning skills progressively during training.

- The authors provide comprehensive open-source resources, demonstrating the reproducibility of the models. These resources are valuable for the LLM community.

**Weaknesses:**

- Missing discussion on the applications. Although the authors conduct experiments and provide insights on training LLMs and improving their reasoning capability, this paper does not discuss how to apply the insights to enhance the LLM products in different domains.

- Unclear construction of training corpus. The author should provide more details about data collection, data cleaning, and training data construction. The authors use fuzzy data deduplication, but they have not explained the tools of fuzzy. They should open-source the data for reproducibility. Besides, the detailed model architecture is missing.

- Table 7 is confusing. The results in Table 7 show the code data will lead to a performance drop on four out of five datasets. It indicates that the code data may not help to improve the reasoning capability of LLMs. The authors should provide valid reasons.

- The related work is limited. Recently, there have been various papers discussing the reasoning capability of LLMs. Therefore, the authors should survey more related papers and compare with them.

- Fix the grammar errors and improve the presentation. On page 8, “The experiment found that…’’ -> “The experiment showed that’’.

- Missing future work. The authors should provide the potential future work on LLMs based on the experimental results and insights provided in this paper.

[1] Roziere B, Gehring J, Gloeckle F, et al. Code llama: Open foundation models for code[J]. arXiv preprint arXiv:2308.12950, 2023.

**Questions:**

Please check in Strengths and Weaknesses.

---

> ### Author Response · Authors · 2023-11-15
> **Response to Reviewer L6FP [1/2]**
>
> Thanks for your valuable and constructive comments. We carefully address your concerns as follows.
>
> ## **Applications**
> The main contribution of this paper is to explore the impact of code data on LLM reasoning capabilities at different training stages and draw the following conclusions. Firstly, adding code data in the pre-training stage can effectively improve the model's general reasoning capabilities. Secondly, adding code instructions in the instruction fine-tuning stage can improve specific code reasoning capabilities. Moreover, the dynamic mixing strategy of code and text data assists LLMs in learning reasoning capability step-by-step during training. Below, we provide some ideas for further applications of these conclusions. We have added these discussions to the revised paper and highlighted them.
>
>
> + **Improve General Reasoning Skills.** The conclusions obtained in this article have direct implications for the application of general LLMs in multiple reasoning-intensive fields (such as legal support, scientific question answering, etc.). Secondly, for the construction of large models in vertical fields (such as large legal models, etc.), it is often necessary to use vertical field data for continued pre-training. High-quality data is the key to model training. If there is a lack of data in vertical fields, you can try to mix certain high-quality code data to improve the model's reasoning capabilities in vertical fields.
>
>
> + **Improve Domain-specific Reasoning Skills.** This paper finds that mixing in code data during the fine-tuning phase can improve capabilities in specific areas of the code. This may be because instructions for related tasks may activate reasoning and response abilities in the corresponding tasks. Therefore, for specific fields, you can also consider mixing in instructions from specific fields to further improve reasoning capabilities.
>
>
> + **Importance of Training Data.** This paper verifies the impact of code data through ablation experiments and proves the importance of data. However, there are still data that may be similar to code data, such as scientific paper data, mathematical data, etc. This may also be the key to improving the reasoning ability of language models. The experimental exploration in this article provides a direction for further research on how multiple types of data affect model training.
>
> ## **Training Corpus**
> + **Data Collection**. The data for pre-training is collected from public datasets, including BaiDuQA, CAIL2018, SogouCA, and CodeParrot, and crawled data, including Common Crawl, Encyclopedias, News, and e-books. The constructed corpus contains about 42B tokes.
>
>
> + **Data Cleaning**. We use the rule-based data cleaning and model-based data filtering methods to ensure the quality of the data. More details can be found in Section F in the Appendix of the original paper. Regarding the cleaning of code data, you can refer to the **Quality of Code Data** section in the reply to the Reviewer PRkj.
>
>
> + **Tool of Fuzzy**. For fuzzy data deduplication, we employed Spark's MinHashLSH algorithm [1], which is a widely adopted technique in advanced LLMs such as GPT-3. We have revised our paper to expound on these specifics, and the revised parts have been highlighted.
>
>       [1] https://spark.apache.org/docs/3.1.1/api/python/reference/api/pyspark.ml.feature.MinHashLSH.html
>
>
> + **Data Mixing Strategy**. The mixing ratios of different data resources are demonstrated in Figure 2 of our original paper. Concretely, the ratios of CodeParrot, Common Crawl, e-Books, News, Encyclopedias, and Public Datasets are 36.1\%, 6.00\%, 13.50\%, 13.50\%, 14.00\%, and 16.90\%, respectively. Besides, to ensure that the model is not biased towards any specific data type during the pre-training process, we randomly shuffle the samples in the training corpus. In addition, during the instruction-tuning phase, we discuss the impact of different data-mixing strategies on model performance. For details, please refer to Section 3.3.4 of the original article.
>
> + **Open-souce Data**. Thanks for your constructive suggestion. Most of the components of our training corpus are open-source datasets. We will consider releasing our training corpus to guarantee reproducibility and make a data contribution if the paper is accepted.

---

> ### Author Response · Authors · 2023-11-15
> **Response to Reviewer L6FP [2/2]**
>
> ## **Detailed Model Architecture**
> Our model, along with other LLMs like Llama and Google PaLM, shares a common architecture based on the GPT-3 causal decoder architecture. Besides, the pre-training task is the next token prediction task. Furthermore, we refer to the structural design of open-source models such as PanGu-alpha, CPM2.6B, and EVA2.6B to retain the setting of the 32-layer transformer decoder. The detailed model architecture diagram is shown in Appendix B.
>
>
>
> ## **Table 7**
> Our paper extensively evaluates various reasoning tasks, including logical and code reasoning, highlighting the positive impact of code-related data. Additionally, we sought to ascertain whether code data would affect common-sense tasks (which do not require high reasoning abilities). To address this, we conducted experiments outlined in Table 7, revealing that code data had minimal influence on performance across other tasks.
>
>
>
> ## **Related Work**
> Thanks for your comment. We have added the following discussion to related work in our revised paper. This part has been highlighted in the paper. HELM test [2] provides a large-scale evaluation of existing large language models. They found that the code-cushman-001model trained on code data had more reasoning power than other natural language models. The AI2 [3] work also shows that, when equipped with complex thought chains, the code-davinci-002 model is currently the best-performing model on important mathematical benchmarks such as GSM8K. In addition, some work analyzed the reasoning ability of large language models through the way of thinking chain and found that because the original GPT-3 had not been subjected the code training, it could not do CoT reasoning [4]. The PaLM training data contains 5% of the code training data, which shows that PaLM can effectively perform CoT reasoning. However, in the above work, the evaluated models have different parameters and data sizes, and there are problems, such as unknown training details. Therefore, it is not rigorous to speculate the exact impact of code data on reasoning ability only by comparing existing models.
>
>     [2] Liang P, Bommasani R, Lee T, et al. Holistic evaluation of language models[J]. arXiv preprint arXiv:2211.09110, 2022.
>
>     [3] Fu Y, Peng H, Sabharwal A, et al. Complexity-based prompting for multi-step reasoning[J]. arXiv preprint arXiv:2210.00720, 2022.
>
>     [4] Wei J, Wang X, Schuurmans D, et al. Chain-of-thought prompting elicits reasoning in large language models[J]. Advances in Neural Information Processing Systems, 2022, 35: 24824-24837.
>
>
>
> ## **Grammar Errors**
> Thank you for your comments on the expression of the paper. We have corrected this grammar error in the revised paper. We will carefully correct the typos and further improve the presentation in the future.
>
>
>
> ## **Future Work**
> Models such as ChatGPT illustrate that larger-scale models will produce emergent capabilities. Therefore, the follow-up of this paper will study how to code the impact of data on models of different sizes and explore the relationship between code data and emergent capabilities. We have added these discussions to the revised paper and highlighted them.

---

> ### Comment · Reviewer_L6FP · 2023-11-21
> **Respond to the Authors**
>
> Thanks for the clarification, my concerns have been addressed. After reading the responses of the authors and other reviews, I decided to raise my score. The reasons are as follows. The authors add a discussion of applications, details of dataset construction, and related work. Other reviewers also considered the paper to be a valuable research work.

---

> > ### Author Response · Authors · 2023-11-21
> > **Thanks to Reviewer L6FP**
> >
> > Thanks for your professional comments, which significantly improved the quality and comprehensiveness of this paper. We also thank you very much for your recognition of this work, and we will further improve this research in the future.

---

### Official Review · Reviewer_PRkj · 2023-10-29

**Soundness:** 4 excellent
**Presentation:** 3 good
**Contribution:** 3 good
**Rating:** 8
**Confidence:** 5

**Summary:**

This paper aims to answer an important research question: at which training stage does code data help LLMs reasoning? The authors introduce code at the pre-training stage, instruction-tuning stage, and both. The reasoning capability of LLMs is evaluated by six reasoning tasks. Through comprehensive experiments and careful analyses, they provide inspiring conclusions and insights. The authors open-source the code and model parameters.

**Strengths:**

1. Valuable research question. The paper raises a meaningful research question: at which training stage introducing code data can really help the reasoning capabilities of LLM? This question is of critical significance for understanding the training and application of LLM.

2. Comprehensive experimental design. This paper provides a comprehensive and fair evaluation of the reasoning capabilities of LLMs on six reasoning tasks covering five domains. This broad experimental scope ensures the generalizability and reliability of the conclusions. Additionally, the authors compare models with different sizes to verify the generalization of the conclusion.

3. In-depth analyses and insights. The paper not only provides experimental results but also performs in-depth analysis, providing insights into mixing code and text data to enhance the general reasoning capabilities and code reasoning capabilities of LLM. Specifically, in the pre-training stage, mixed code data helps LLM improve general reasoning capabilities, and in the SFT stage, mixed code data helps LLM improve specific code reasoning capabilities.

**Weaknesses:**

1. Experimental details are insufficient. The paper may not provide enough details on experimental settings and parameter selection in some parts (such as data mixing strategies, decoding strategies, etc.). This might challenge researchers attempting to replicate or extend this work.

2. Recently, various code foundation models, such as CodeLlama [1], have been opened. However, the authors do not conduct any discussions or experiments on them. In my opinion, the reasoning capability of code foundation models is also an essential part of the interest scope of this work.

3. Quality of code data. The quality of code data can have a significant impact on the reasoning capabilities of LLM. How the author ensures the high quality of code data is not discussed in depth in the article.

4. The related work part is weak. Missing important papers, such as [1,2,3].
[1] Roziere B, Gehring J, Gloeckle F, et al. Code llama: Open foundation models for code[J]. arXiv preprint arXiv:2308.12950, 2023.
[2] Yang A, Xiao B, Wang B, et al. Baichuan 2: Open large-scale language models[J]. arXiv preprint arXiv:2309.10305, 2023.
[3] Li P, Sun T, Tang Q, et al. CodeIE: Large Code Generation Models are Better Few-Shot Information Extractors[J]. arXiv preprint arXiv:2305.05711, 2023.

**Questions:**

1. Experimental details. In the paper, certain key sections, such as data mixing strategy and decoding strategy, do not seem to give sufficient details of experimental settings and parameter selection. This lack of information can lead to difficulties in reproducing and scaling. How was it set up by the author in the actual experiment?

2. Quality of code data. How were the coding data used by the authors in their experiments collected and screened? The quality of code data can have a significant impact on the reasoning capabilities of LLM. For example, high-quality code data may provide more reasoning impact, while low-quality data may cause the model to learn wrong patterns. How did the authors ensure that the code data used were representative and of high quality?

---

> ### Author Response · Authors · 2023-11-13
> **Response to Reviewer PRkj [1/2]**
>
> Thanks for your valuable and constructive comments. We carefully address your concerns as follows.
>
> ## **Experimental Details**
> + **Data Resources**. The data for pre-training is collected from public datasets, including BaiDuQA, CAIL2018, SogouCA, and CodeParrot, and crawled data, including Common Crawl, Encyclopedias, News, and e-books. The constructed corpus contains about 42B tokes.
>
> + **Data Cleaning**. We use the rule-based data cleaning and model-based data filtering methods to ensure the quality of the data. More details can be found in Section F in the Appendix of the original paper.
>
> + **Data Mixing Strategy**. The mixing ratios of different data resources are demonstrated in Figure 2 of our original paper. Concretely, the ratios of CodeParrot, Common Crawl, e-Books, News, Encyclopedias, and Public Datasets are 36.1\%, 6.00\%, 13.50\%, 13.50\%, 14.00\%, and 16.90\%, respectively. Besides, to ensure that the model is not biased towards any specific data type during the pre-training process, we randomly shuffle the samples in the training corpus. In addition, during the instruction-tuning phase, we discuss the impact of different data mixing strategies on model performance. For details, please refer to Section 3.3.4 of the original article.
>
> + **Decoding Strategy**. For the generation task, we adopt a unified decoding strategy for all models, namely, the top-k sampling strategy, where k equals 3. Besides, for the classification task, we adopt a perplexity-based greedy decoding method. Concretely, the category with the minimum perplexity value is regarded as the predicted label of the paragraph.
>
> + **Parameters**. The detailed configurations, including environmental parameters, model parameters, optimization parameters, and parallelism parameters are summarized at Table 9 in the Appendix of the original paper.
>
>
> ## **CodeLLama**
> Thank you for sharing, we have noticed the release of Code Llama, a high-performance model focusing on the code field. However, this article focuses on how code data affects the reasoning capabilities of general LLM, while Code Llama focuses on improving the code capabilities of LLM. We carefully read the training process and result evaluation of Code Llama, and also discovered some interesting phenomena.
>
> * **Code Reasoning Ability**. Code Llama continues to pre-train on code data based on Llama 2. We have observed the improvement in code ability brought about by the increase in code data.
>
> * **Mathematical Ability**. We have observed from the open LLM Leaderboard that Code Llama' s training on more code data (non-mathematical data) has also improved mathematical ability. We infer that it may be related to the improvement of LLM's general reasoning ability by code data, or it may be related to the consistency between mathematical data and code data. We will add our discussions to related work, as shown in the figure below.
>
>   | Model            | HumanEval (CODE) | GSM8K (MATH) |
>   |------------------|------------------|--------------|
>   | Llama 2-7B       | 12.2             | 3.49         |
>   | Code Llama-7B    | **33.5**             | **5.16**         |
>   | Llama 2-13B      | 20.1             | 10.84        |
>   | Code Llama-13B   | **36.0**             | **12.13**        |
>
>     *Note: HumanEval scores are from the original Code Llama report, GSM8K scores from the Open LLM Leaderboard(https://huggingface.co/spaces/HuggingFaceH4/open_llm_leaderboard) .*

---

> ### Author Response · Authors · 2023-11-15
> **Response to Reviewer PRkj [2/2]**
>
> ## **Quality of Code Data**
> The code data used in this article mainly comes from CodeParrot's open-source data, which comes from Github's Python repository. This data set is adopted by a wide range of LLMs, such as CodeGeeX [1], InCoder [2], CodeParrot [3], etc. Since the original data set contains a large amount of duplicate data and noise data, to ensure the code data quality, we use the "codeparrot-clean" version, which is the deduplication version of codeparrot. This version filters and deduplicates codeparrot based on the number of lines of code and the proportion of characters and numbers. The specific steps are as follows.
> * Deduplication
>   * Remove exact matches
> * Filtering
>   * Average line length < 100
>   * Maximum line length < 1000
>   * Alpha numeric characters fraction > 0.25
>   * Remove auto-generated files (keyword search)
>
> For detailed processing procedures, please visit the following link:https://huggingface.co/datasets/codeparrot/codeparrot-clean. Finally, ~50GB of code data was obtained for training.
>
>     [1] Zheng Q, Xia X, Zou X, et al. Codegeex: A pre-trained model for code generation with multilingual benchmarking on humaneval-x[C]//Proceedings of the 29th ACM SIGKDD Conference on Knowledge Discovery and Data Mining. 2023: 5673-5684.
>
>     [2] Fried D, Aghajanyan A, Lin J, et al. Incoder: A generative model for code infilling and synthesis[J]. arXiv preprint arXiv:2204.05999, 2022.
>
>     [3] Xu F F, Alon U, Neubig G, et al. A systematic evaluation of large language models of code[C]//Proceedings of the 6th ACM SIGPLAN International Symposium on Machine Programming. 2022: 1-10.
>
> ## **Related Paper**
> Thank you for providing relevant work references [4-6]. We have added these discussions to the revised paper and highlighted them.
>
>     [4] Roziere B, Gehring J, Gloeckle F, et al. Code llama: Open foundation models for code[J]. arXiv preprint arXiv:2308.12950, 2023.
>
>     [5] Yang A, Xiao B, Wang B, et al. Baichuan 2: Open large-scale language models[J]. arXiv preprint arXiv:2309.10305, 2023.
>
>     [6] Li P, Sun T, Tang Q, et al. CodeIE: Large Code Generation Models are Better Few-Shot Information Extractors[J]. arXiv preprint arXiv:2305.05711, 2023.

---

> > ### Comment · Reviewer_PRkj · 2023-11-21
> > **Thanks for the author's response**
> >
> > The author has well addressed most of my concerns. The author has provided a new idea and verification for studying the reasoning ability of large language models, which can inspire other researchers in the community. In addition, I have carefully read the comments of other reviewers and the responses of the authors, so I recommend accepting this paper.

---

> ### Author Response · Authors · 2023-11-21
> **Thanks to Reviewer PRkj**
>
> Thank you for your insightful and constructive feedback, which has greatly enhanced the quality of our paper. We deeply appreciate your positive remarks about our work. Rest assured, we will incorporate your valuable suggestions to further refine our paper for the final version.

---

### Official Review · Reviewer_BTot · 2023-10-31

**Soundness:** 3 good
**Presentation:** 3 good
**Contribution:** 3 good
**Rating:** 8
**Confidence:** 3

**Summary:**

Previous work has shown that models trained on code perform better on reasoning tasks. The research question in this paper is at which training stage code data helps reasoning. Specifically, this work looks into two training stages: pre-training and instruction-tuning. The results show that for performance on reasoning tasks, adding code data at the pre-training stage is effective whereas at the instruction-tuning the effect is far less (sometimes leading to lower performance); however, instruction tuning on code can improve model performance on code-related problems.

**Strengths:**

- **Clear Hypothesis:** The paper asks a clear question and provides a clear setup for testing various hypothesis about that question.
- **Clear message:** The results show a clear message about the research question, albiet only on smaller-sized models.
- **Clarity of the writing:** The writing was mostly clear and easy to follow

**Weaknesses:**

- **Datasets:** I was not familiar with some of the datasets used in this work and after looking into some of them, I could not get a sense of how general and challenging they are. The majority of the computation cost for this project seems to be on the training stage, so I believe reporting results on a few more datasets (maybe only for Tables 2 and 3) can strengthen the main arguments of the paper. That could include datasets from other reasoning domains (e.g., math might be an important one that does not appear in the results) or from the same domains but on datasets that are more established.
- **Mixture experiment:** The experiment on exploring ways to mix code and text data is interesting, but given that adding code at the instruction-tuning stage was already shown to be not that effective, I wonder why it was tested for the instruction-tuning stage. I understand the high computation cost of pre-training, but it seems to me that given the previous set of results, this experiment makes sense mostly at the pre-training stage where we have seen that code data can be effective.
- **Language Inconsistency:** - For the results in Section 3.3.5, the authors conclude that training with code data *has little negative impact* on the performance of other tasks. But the numbers in Table 7 don't seem to show little negative impact. The large impact on the DuReader dataset has been already pointed out by the authors. Moreover, on CMNLI the performance decreases from 45.07 to 43.49 which is almost equal in magnitude to some of the gains reported in Table 2. I believe the language should be more consistent on the amount of improvement/decrement that can be considered a significant amount for the datasets and the experimental setup of the paper.
- **Minor suggestions:** 1- In Table 3, there are multiple equal numbers but only one of them is in bold face. 2- In Table 4, I suggest reversing the rows and columns to make it consistent with the other tables.

**Questions:**

- For the Logic dataset, I see that the accuracy for many different models is 40.9. This value appears in Table 2, Table 3 and Table 6. Does this hint at a potential problem in this dataset?
- Some of the datasets don't seem to be available in English (or at least I could not find an English version of them). Could you include some examples from these datasets translated to English, so the reader can get a sense of the nature and the difficulty level?
- It has been observed that larger LLMs often behave differently than smaller LLMs. I'd be curious to hear the authors' thoughts on how they think their results might transfer to larger models? (to be clear, I understand the computation cost and I'm not suggesting that you run those experiments)

---

> ### Author Response · Authors · 2023-11-17
> **Response to Reviewer BTot [1/3]**
>
> ## **Datasets**
> In terms of performance evaluation, the datasets used in this paper are all public datasets, which are widely adopted by many methods for testing model performance. For example, the Logic data comes from C_Eval [1], and models such as ChatGLM [2] and Qwen [3] have been tested on this dataset. Besides, MBPP [4] is a widely used code testing benchmark and is widely used by models such as GPT-4 [5], Llama 2 [6], etc.
>
>     [1] Huang Y, Bai Y, Zhu Z, et al. C-eval: A multi-level multi-discipline chinese evaluation suite for foundation models[J]. arXiv preprint arXiv:2305.08322, 2023.
>
>     [2] Zeng A, Liu X, Du Z, et al. Glm-130b: An open bilingual pre-trained model[J]. arXiv preprint arXiv:2210.02414, 2022.
>
>     [3] Bai J, Bai S, Chu Y, et al. Qwen technical report[J]. arXiv preprint arXiv:2309.16609, 2023.
>
>     [4] Austin J, Odena A, Nye M, et al. Program synthesis with large language models[J]. arXiv preprint arXiv:2108.07732, 2021.
>
>     [5] https://openai.com/research/gpt-4
>
>     [6] Touvron H, Martin L, Stone K, et al. Llama 2: Open foundation and fine-tuned chat models[J]. arXiv preprint arXiv:2307.09288, 2023.
>
>
> + **General and Challenging Datasets**.
>
>   + **For the general test**, to comprehensively test the reasoning abilities of the models, we used different types of reasoning tasks and datasets for performance testing, including logical reasoning, legal reasoning, scientific question and answer, and code generation. These tasks comprehensively reflect the model's performance on different types of reasoning capabilities.
>
>   + **For the challenge**, the evaluated models need to achieve promising reasoning results on all types of reasoning tasks, which is very challenging. Besides, the datasets used in this article usually require multi-step reasoning to obtain correct results, further improving the difficulty of reasoning.
>
>
> + **Examples of Datasets**.
> Sorry for causing trouble for your understanding. For the non-English data sets used in this article, following your suggestion, we have translated them into English for display. Below are examples from the Logic, JEC-QA, and E-KAR datasets. We have added these examples to the revised paper and highlighted them.
>
>   + **One Example of Logic (Logical Reasoning)**
>
>         Problem:
>
>         Regarding the physical education standard test for Class A, three teachers made the following predictions: Teacher Zhang said, "Not everyone will fail." Teacher Li said, "Someone will fail." Teacher Wang said, "Both the class president and the study committee member will pass." If only one of these teachers' predictions is correct, which of the following must be true?
>
>         Answer List:
>
>         A: "Both the class president and the study committee member failed."
>         B: "Both the class president and the study committee member passed."
>         C: "The class president passed, but the study committee member failed."
>         D: "The class president failed, but the study committee member passed."
>
>         Answer: A
>
>   + **One Example of JEC-QA (Legal QA)**
>
>
>         Problem:
>
>         A miscellaneous article written by person A caused a significant stir after its publication. The article was reprinted by several newspapers and websites without compensation. Person B translated the article into French and person C translated it into Uighur. Both translations were published domestically without A's consent and without any remuneration. Which of the following viewpoints is correct?
>
>         Answer List:
>
>         A: "The act of newspapers and websites reprinting the article does not constitute infringement."
>         B: "The actions of both B and C do not constitute infringement."
>         C: "B's action does not constitute infringement, but C's action does."
>         D: "B's action constitutes infringement, but C's action does not."
>
>         Answer: C
>
>
>
>   + **One Example of E-KAR (Analogical Reasoning)**
>
>         Problem:
>
>         Based on the given relationship [Speed:Time:Distance], choose the option that fits this relationship.
>
>         Answer List:
>
>         A: "Interest Rate:Principal:Interest"
>         B: "Quality:Variety:Quantity"
>         C: "Profit:Cost:Value"
>         D: "Income:Expenditure:Surplus"
>
>         Answer: A
>
>         Explanation: From the given relationship, we infer the following: "Speed" multiplied by "Time" equals "Distance". In option A, "Interest Rate" multiplied by "Principal" equals "Interest". In option B, there is no clear logic connecting "Quality", "Variety", and "Quantity". In option C, the product of "Profit" and "Cost" is not "Value". In option D, "Surplus" is the difference between "Income" and "Expenditure". Therefore, the correct choice is A.

---

> ### Author Response · Authors · 2023-11-17
> **Response to Reviewer BTot [2/3]**
>
> ## **Datasets**
> + **Other Datasets**.
>
>   In order to more comprehensively verify the observations of this article, we selected the high school mathematics and high school physics problem parts of the MMLU[1] test set to evaluate the model in the pre-training stage. MMLU is a currently widely used dataset to evaluate the comprehensive ability of LLMs [2, 3], among which mathematics and physics can better reflect the reasoning ability of the model. The results obtained and the conclusions are shown in the table below. We have added these results to the revised paper and highlighted them. Thanks for your suggestions again.
>
>       [1] Hendrycks D, Burns C, Basart S, et al. Measuring massive multitask language understanding[J]. arXiv preprint arXiv:2009.03300, 2020.
>
>       [2] https://openai.com/research/gpt-4
>
>       [3] Touvron H, Martin L, Stone K, et al. Llama 2: Open foundation and fine-tuned chat models[J]. arXiv preprint arXiv:2307.09288, 2023.
>
>
>
>   | Task       | NL (2.6B) | NL (13B) | **CODE (2.6B)** | p value| *LLaMA (7B)*  |
>   |------------|--------|-------|--------------|------------------------------------------|---------|
>   | MMLU_Math  | 24.16*  | 22.30 | **24.91**   | <0.05    |  *24.97*  |
>   | MMLU_Physics| 20.00 | 22.67* | **26.67**    | <0.05    |  *27.97*  |
>
>   *Note: Source of LLaMA results: https://github.com/baichuan-inc/Baichuan-7B*
>
>   In this table, **bold** values denote the best result, and the values* denote the runner-ups in the scope of this paper. The t-test is conducted to verify the significant improvement of the best values compared with the runner-up values. Besides, the italics denotes the models out of the scope of this paper. From these experimental results, we have three conclusions as follows.
>
>   + On mathematical and physical reasoning tasks, the CODE (2.6B) model shows advantages over NL (2.6B) and NL (13B), which strengthens the effectiveness of introducing code data in the pre-training stage. We performed a t-test for statistical significance, and the results showed that the p-value was less than 0.05, which indicated that the results were statistically significant.
>   + We admit that models, including LLaMA (7B) and CODE (2.6B), perform relatively poorly on mathematical and physical tasks, but we believe that the relative improvement brought by code data is trustworthy. This also shows how it is necessary to further improve mathematics-related tasks.
>
>
> + **Logic Dateset**.
> Thanks for your reminder. We carefully inspected the Logic dataset and the output of models. We found that the distribution of the predicted scores is different between different models. Besides, some of the answers to the questions overlapped while others were different. The similar low accuracy might be due to the difficulty of the Logic dataset.
>
>
> ## **Mixture Experiment**
> As you said, due to resource constraints, we only conducted data mixing experiments during the instruction-tuning phase. In this experiment, we found that using a higher proportion of code data in the early stage can improve the specific code reasoning ability of the language model. We think this may have implications for some vertical scenarios that require fine-tuning to implement. At the same time, in view of the findings in the fine-tuning phase, we are also planning to explore data mixing experiments in the pre-training phase. We believe this is a promising direction. Thanks for your suggestion.
>
> ## **Language Inconsistency**
> Thanks for your suggestion. After carefully verifying the results of our article, particularly in section 3.3.5, we found that the code data had an impact on the DuReader task. We believe this may be due to the model not fully learning from the code and text data, leading to confusion when generating answers to reading comprehension questions. To avoid misleading the readers, we have improved our conclusion in the revised paper to state: "Using code data for training may negatively impact the performance of other tasks."
>
> ## **Minor Suggestions**
> Thanks for your suggestions.
> - Regarding the bolding of identical performances, we have made the corrections in the revised paper.
> - For Table 4, following your suggestion, we have reversed the rows and columns to make it consistent with other tables.

---

> > ### Author Response · Authors · 2023-11-17
> > **Response to Reviewer BTot [3/3]**
> >
> > ## **Larger LLMs**
> > Thank you for your question. We acknowledge that the current study does not cover larger-scale models, but this is an important direction for future research.
> >
> > In the future, we plan to study the **scaling law** of large language models (LLM) on **code data**. There are various famous scaling laws of the general LLMs, such as KM scaling law [1] and Chinchilla scaling law [2]. Besides, GPT-4 introduces a new mechanism called predictable scaling, enabling the performance prediction of LLMs with much smaller models. Scaling Laws state that the performance of a model improves as the size of the model, the size of the data set, and the number of computational floats used for training increase. And for optimal performance, all three factors must be amplified simultaneously. When not constrained by the other two factors, model performance has a power-law relationship with each individual factor. Although promising processes have been made, the scaling law for code LLMs is still an open question.
> >
> >     [1] Kaplan J, McCandlish S, Henighan T, et al. Scaling laws for neural language models[J]. arXiv preprint arXiv:2001.08361, 2020.
> >     [2] Hoffmann J, Borgeaud S, Mensch A, et al. Training compute-optimal large language models[J]. arXiv preprint arXiv:2203.15556, 2022.
> >     [3] OpenAI R. Gpt-4 technical report. arxiv 2303.08774[J]. View in Article, 2023, 2.
> >
> > In subsequent experiments, we plan to focus on **exploring how the amount of code data, model complexity, and computing resources affect the reasoning performance** of large-scale models. Analyze the performance of models of different sizes on various inference tasks and look for the relationship between model size, amount of training data (especially code data), computing resources, and performance. By studying the Scaling Laws of LLM when processing code data it will provide valuable experience and training inspiration for future research.

---

> > > ### Comment · Reviewer_BTot · 2023-11-20
> > > **Thanks**
> > >
> > > Thanks for the detailed responses and for the new experiments and clarifications. I have increased my score.

---

> > > > ### Author Response · Authors · 2023-11-21
> > > > **Thanks to Reviewer BTot**
> > > >
> > > > Thanks for your insightful and constructive reviews, which significantly improve the quality of this paper! Thank very much for appreciating our paper! We will make it better in the final version.

---

### Official Review · Reviewer_viag · 2023-11-01

**Soundness:** 2 fair
**Presentation:** 3 good
**Contribution:** 2 fair
**Rating:** 5
**Confidence:** 4

**Summary:**

The paper explores the effect of including code data during pre-training and instruction tuning over the reasoning capabilities of LLMs. The authors conduct a set of ablation experiments where code is added/removed from both pre-training and fine-tuning and the model performance is measured over a set of reasoning tasks ranging from logical to legal reasoning. The results show that including code data during pre-training is more effective than during instruction tuning. Also, the results show that instruction tuning over the code can end up hurting performance on some tasks. The authors also experimented with a dynamic text-code mixing strategy during instruction tuning.

**Strengths:**

* The problem studied is interesting: the relationship between code data and reasoning and the paper aims to somehow tackle this issue.
* The paper is well-written and the results are well-presented.

**Weaknesses:**

* From my understanding (and correct me If I'm wrong), the code model is trained on more overall tokens than the NL model. I would expect a study like that to control for the number of pre-training tokens while changing their nature i.e., text vs. code. If the code model is trained on as many natural text tokens as the baseline model in addition to having code in the pre-training data, then the code model should be expected to perform better because it was trained on more data. No surprise there.
* It's hard to say whether the results reported have statistical significance. For example, in Table 2, the code model is only 0.13 points better than the NL model on ScienceQA. Are these results significant? And are these enough to conclude that code reasoning? I would expect the authors to run a statistical significance test to support their results.
* The proposed dynamic mixing strategy produces very marginal improvements (except over logical reasoning) and has a negative effect on the performance over three tasks. The paper does not thoroughly investigate why this is the case. Also, the design of the mixing strategy seems rather arbitrary.
* The evaluation does not cover mathematical reasoning, although it's one type of reasoning where we should expect great improvements since code data is roughly similar to math data.

**Questions:**

* The choice of the baseline model is unclear. Why use PanGu2.6B as your baseline model?
* Why does your mixing strategy follow the 7:3, then 6:4, then 5:5? Why not, for example, 9:1, 7:3, 5:5, 3:7, 1:9? which would be both increasing and decreasing. And why only 4 phases? What's the intuition behind that design?



===== POST REBUTTAL =====

I thank the authors for their response. Based on the response and the expectation that the authors will add results from training a model on 150G of natural data, I have decided to increase my score to 5.

---

> ### Author Response · Authors · 2023-11-18
> **Response to Reviewer viag [1/3]**
>
> ## **Training Data**
> Thanks for your insightful and constructive comments.
>
> + **Training Corpus**. Your understanding is partly correct. Actually, NL (2.6B) is trained on about 100G natural language data. Besides, CODE (2.6B) is trained on about 100G natural language data plus 50G code data. In addition, NL (13B) is trained on more than 150G natural language data.
>
> + **Effectiveness**. In the pre-training and instruction tuning stages, we compare the above three models, including NL(2.6B), NL(13B), and CODE(2.6B). In the pre-training stage, Table 2 shows that, on the reasoning task, CODE (2.6B) with mixed code data and natural language not only outperforms NL (2.6B) with natural language but also outperforms the larger pure text model NL (13B) trained on more text data. These experimental results highlight the importance of code data in the pre-training stage. Besides, at the instruction tuning stage, Table 3 demonstrates that CC (2.6B), which is fine-tuned on code instructions, performs better on specific code tasks, highlighting the importance of code data in the instruction tuning stage.
>
> + **Additional Study**. Thanks for your insightful suggestion! We have already begun our pre-training of NL(2.6) on 150G natural language data. Due to the limitation of computational resources, it will take about 15 days. We will add the results and conclusions in the revised paper to further improve the comprehensiveness and quality of this paper.
>
> + **More Data**. Actually, as claimed in **Effectiveness**, the fact that CODE (2.6B) beats NL (13B) on reasoning can show the importance of code data in improving the reasoning capacity of models. Besides, we do not agree that simply adding more data can definitely result in better performance. The reasons are as follows. 1) Large amounts of data may easily lead to underfitting problems of the models. 2) The low-quality data may lead to the hallucination problem in LLMs. 3) According to many scaling laws [1, 2, 3], the model size, dataset size, and the amount of training computation should be amplified simultaneously to achieve better performance. Therefore, according to the extensive experiments and analyses as claimed in **Effectiveness**, we demonstrate the importance of the code data in both pre-training and fine-tuning of LLMs.
>
>       [1] Kaplan J, McCandlish S, Henighan T, et al. Scaling laws for neural language models[J]. arXiv preprint arXiv:2001.08361, 2020.
>
>       [2] Hoffmann J, Borgeaud S, Mensch A, et al. Training compute-optimal large language models[J]. arXiv preprint arXiv:2203.15556, 2022.
>
>       [3] OpenAI R. Gpt-4 technical report. arxiv 2303.08774[J]. View in Article, 2023, 2.
>
> ## **t-Test**
> Thanks.
> + In order to comprehensively evaluate the model's reasoning ability, we used different types of reasoning tasks and datasets for testing, including logical reasoning, legal reasoning, scientific question and answer, and code generation. These tasks comprehensively reflect the model's performance on different types of reasoning capabilities. Due to the diversity and complexity of these reasoning tasks, achieving promising performance on all tasks is a challenge. In Table 2, compared with NL (2.6B), we find that CODE (2.6B) can bring 4.54% and 3.88% performance improvement on MBPP and E-KAR datasets.
>
> + Besides, following your suggestion, in order to illustrate the significance of these results, we conducted a t-test on the predicted scores in the following table. It demonstrates that all p-values are less than 0.05, indicating that the results are statistically significant. We have already add these results to the revised paper and highlighted them.
>
>   | Dataset | p-value (<0.05) |
>   |-----------|-----------------|
>   | Logic     | 4.197e-06       |
>   | JEC-QA    | 1.956e-25       |
>   | ScienceQA | 0.014          |
>   | E-KAR     | 7.013e-07       |
>   | CosQA     | 1.066e-40       |

---

> ### Author Response · Authors · 2023-11-18
> **Response to Reviewer viag [2/3]**
>
> ## **Model Choice**
> - PanGu, along with other LLMs like Llama and Google PaLM, shares a common architecture based on the GPT-2 decoder-only architecture and next token prediction task. In addition, due to resource limitations, we selected the 2.6B model for experiments and compared it with the 13B model. Because open-source models such as CPM2.6B [3], EVA2.6B [4], and ChatGLM2.6B [5] all use the 2.6B scale and show that they already have some general capabilities.
>
> - At the same time, we look forward to the emergence of LLMs with different architectures than GPT, and we also look forward to following up and verifying more LLMs with different architectures.
>
> - In this research, the GPU and funding supporters adopt the PanGu model. Similar facts are that Google prefers PaLM, OpenAI prefers GPT, and Meta prefers LLaMA. We focus more on exploring the research question itself, i.e., at which stage can code data help LLMs reasoning. The model choice should not be an advantage or a disadvantage.
>
>       [3] Zhang Z, Han X, Zhou H, et al. CPM: A large-scale generative Chinese pre-trained language model[J]. AI Open, 2021, 2: 93-99.
>
>       [4] Zhou H, Ke P, Zhang Z, et al. Eva: An open-domain chinese dialogue system with large-scale generative pre-training[J]. arXiv preprint arXiv:2108.01547, 2021.
>
>       [5] https://github.com/THUDM/ChatGLM2-6B
>
> ## **Mathematical Reasoning**
>
> Following your suggestion, we selected the high school mathematics and high school physics problem parts of the MMLU [6] test set to evaluate the model in the pre-training stage. MMLU is a currently widely used dataset to evaluate the comprehensive ability of LLMs [7, 8], among which mathematics and physics can better reflect the reasoning ability of the model. The results obtained and the conclusions are shown in the table below. We have added these results to the revised paper and highlighted them. Thanks for your suggestions again.
>
>     [6] Hendrycks D, Burns C, Basart S, et al. Measuring massive multitask language understanding[J]. arXiv preprint arXiv:2009.03300, 2020.
>
>     [7] https://openai.com/research/gpt-4
>
>     [8] Touvron H, Martin L, Stone K, et al. Llama 2: Open foundation and fine-tuned chat models[J]. arXiv preprint arXiv:2307.09288, 2023.
>
>
>
>   | Task       | NL (2.6B) | NL (13B) | **CODE (2.6B)** | p value| *LLaMA (7B)*  |
>   |------------|--------|-------|--------------|------------------------------------------|---------|
>   | MMLU_Math  | 24.16*  | 22.30 | **24.91**   | <0.05    |  *24.97*  |
>   | MMLU_Physics| 20.00 | 22.67* | **26.67**    | <0.05    |  *27.97*  |
>
>   *Note: Source of LLaMA results: https://github.com/baichuan-inc/Baichuan-7B*
>
>   In this table, **bold** values denote the best result, and the values* denote the runner-ups in the scope of this paper. The t-test is conducted to verify the significant improvement of the best values compared with the runner-up values. Besides, the italics denotes the models out of the scope of this paper. From these experimental results, we have three conclusions as follows.
>
>   + On mathematical and physical reasoning tasks, the CODE (2.6B) model shows advantages over NL (2.6B) and NL (13B), which strengthens the effectiveness of introducing code data in the pre-training stage. We performed a t-test for statistical significance, and the results showed that the p-value was less than 0.05, which indicated that the results were statistically significant.
>   + We admit that models, including LLaMA (7B) and CODE (2.6B), perform relatively poorly on mathematical and physical tasks, but we believe that the relative improvement brought by code data is trustworthy. This also shows how it is necessary to further improve mathematics-related tasks.

---

> ### Author Response · Authors · 2023-11-18
> **Response to Reviewer viag [3/3]**
>
> ## **Dynamic Mixing Strategy**
> Thanks for your concern, which can help us to improve the paper.
>
> - In order to comprehensively evaluate the model's reasoning ability, we used different types of reasoning tasks and datasets for testing, including logical reasoning, legal reasoning, scientific question and answer, and code generation. These tasks comprehensively reflect the model's performance on different types of reasoning capabilities. Due to the diversity and complexity of these reasoning tasks, achieving promising performance on all tasks is a challenge. We admit that the dynamic mixing strategy can not achieve the best results on all reasoning tasks. But it is really an interesting idea that can improve the reasoning capacity of LLMs during training. We have not carefully studied the segment setting and mixing ratios. The experimental setting is a preliminary attempt. Therefore, further improvements can be made in the future, such as setting up more segments and analyzing different ratios. We believe this research question will be a promising direction.
>
> - Still, under these settings, we obtained some interesting and insightful experimental conclusions in the original paper, especially on the coding task. We have added these details in the revised paper and highlighted them.
>   - Using a larger code proportion in the early stage can improve the performance of LLM in coding tasks (CosQA and MBPP). The reason may be that a higher code proportion in the early stage can better activate code-related reasoning capabilities under a higher learning rate.
>   - The descending strategy can improve the performance of logic. Since the code data is more logical, giving more codes in the initial stage may improve the performance of logic reasoning.
>   - In the other three datasets, uniform is better, probably because these tasks require both logical reasoning ability and common sense as well as natural language understanding ability.
>  Therefore, we recommend choosing different hybrid strategies based on the characteristics of different downstream tasks.
>
> - Designs
>   - The experimental settings of the proposed dynamic mixing strategy are a preliminary exploration. We divided the experiment into four stages based on experience and the proportion of the dataset. Based on your suggestions, we will carefully explore more training stages in the future. We believe it will be a promising and useful training strategy.
>   - In order to maintain the ability of the general model, we do not set widely different stage proportions. Because a ratio that is too high or too low may cause the model to turn into a proprietary code model or a text model. Among them, the 5:3 ratio in the uniform sampling stage is the original text and code instruction data ratio.

---

> > ### Author Response · Authors · 2023-11-21
> > **Follow Up**
> >
> > Dear Reviewer viag,
> >
> > We highly appreciate your valuable and insightful comments. We hope the above response has addressed your concerns. If you have any other suggestions or questions, feel free to discuss them. We are very willing to discuss them with you in this period. If your concerns have been addressed, would you please consider raising the score? It is very important for us and this research. Thanks again for your professional comments and valuable time!
> >
> > Best wishes,
> >
> > Authors

---

### Author Response · Authors · 2023-11-22
**Summary**

We sincerely appreciate your time and effort in reviewing our paper and providing us with valuable feedback. We have carefully considered your suggestions and addressed them in our revised paper. Below, we summarize the main strengths, suggestions for improvement raised by the reviewers, and our solutions.

## **Main Strengths**

+ **Research Problem**: "The problem studied is interesting: the relationship between code data and reasoning, and the paper aims to somehow tackle this issue." "The paper asks a clear question." "Valuable research question. The paper raises a meaningful research question: at which training stage introducing code data can really help the reasoning capabilities of LLM? This question is of critical significance for understanding the training and application of LLM." (Reviewer viag & Reviewer BTot & Reviewer PRkj)

+ **Motivation**: "This paper is well-motivated. The impact of code data in LLMs is a hot research question. This paper answers this issue from the reasoning capability aspect." (Reviewer L6FP)

+ **Writing**: "The paper is well-written, and the results are well-presented." "Clear message: the results show a clear message about the research question." "Clarity of the writing: the writing was mostly clear and easy to follow." (Reviewer viag & Reviewer BTot)

+ **Hypothesis**: "Clear hypothesis: the paper provides a clear setup for testing various hypotheses about that question." (Reviewer BTot)

+ **Experiments**: "Comprehensive experimental design. This paper provides a comprehensive and fair evaluation of the reasoning capabilities of LLMs on six reasoning tasks covering five domains. This broad experimental scope ensures the generalizability and reliability of the conclusions. Additionally, the authors compare models with different sizes to verify the generalization of the conclusion." "The experiments are comprehensive, and the insights are remarkable. The reasoning capability of LLMs is evaluated via six tasks in five domains. The authors provide critical analyses and significant insights on training LLMs and the reasoning capability of LLMs." (Reviewer PRkj & Reviewer L6FP)

+ **Insights and Analyses**: "In-depth analyses and insights. The paper not only provides experimental results but also performs in-depth analysis, providing insights into mixing code and text data to enhance the general reasoning capabilities and code reasoning capabilities of LLM. Specifically, in the pre-training stage, mixed code data helps LLM improve general reasoning capabilities, and in the SFT stage, mixed code data helps LLM improve specific code reasoning capabilities." (Reviewer PRkj)

+ **Idea**: "The idea of dynamic mixed strategy is easy to follow yet effective. It helps LLMs learn reasoning skills progressively during training." (Reviewers L6FP)

+ **Open Resources**: The authors provide comprehensive open-source resources, demonstrating the reproducibility of the models. These resources are valuable for the LLM community. (Reviewers L6FP)

## **Suggestions**

+ Further explanation on the composition and construction of pre-training data (Reviewer viag, BTot, PRkj, and L6FP)
+ Further experiments on mathematical reasoning (Reviewer viag and BTot) and statistical significance (Reviewer viag)
+ Further discussions of the dynamic mixing strategy (Reviewers viag, BTot, and PRkj) and related research (Reviewers PRkj and L6FP)
+ Further discussion regarding the choice of the model (Reviewers viag) and comparisons with larger and other models (Reviewers BTot and PRkj)

## **Solutions**

To address the reviewers' comments, we have taken the following steps:

+ Clarified issues related to the composition and construction of pre-training data.
+ Conducted new additional experiments on mathematical reasoning and statistical significance.
+ Provided a detailed discussion of the dynamic mixing strategy.
+ Provided a detailed discussion of the choice of the model and how to scale to larger models.
+ Added further discussion of related work, future work, and applications for enhanced clarity and rigor.

We have carefully responded to each reviewer's comments. In addition, we have substantially revised the paper and appendix in response to the reviewers' comments. The changed or new sentences in the revision are highlighted in red. Most reviewers have participated in the discussion, and most of their concerns have been addressed. After the discussion, **two reviewers increased their scores in favor of accepting this paper**, and **another reviewer maintained their score and suggested its acceptance**. Once again, we sincerely thank you for your valuable comments and feedback, which significantly improve the quality of this paper.

---

### Meta-Review · Area_Chair_jXwK · 2023-12-14

**Metareview:**

This paper aims to answer an important research question: at which training stage does code data help LLMs reasoning? The authors introduce code at the pre-training stage, instruction-tuning stage, and both. The reasoning capability of LLMs is evaluated by six reasoning tasks. Through comprehensive experiments and careful analyses, they provide inspiring conclusions and insights. The authors open-source the code and model parameters.

The paper was positively received by all reviewers, most note that the analysis and writing is very clear. The authors did a thorough discussion phase. I recommend to accept the paper for Spotlight.

**Justification For Why Not Higher Score:**

Scope

**Justification For Why Not Lower Score:**

N/A

---

### Decision · Program_Chairs · 2024-01-16

Accept (spotlight)